# EMPhone: Electromagnetic Covert Channel via Silent Audio Playback on Smartphones

**DOI:** 10.3390/s25185900

**Published:** 2025-09-21

**Authors:** Yongjae Kim, Hyeonjun An, Dong-Guk Han

**Affiliations:** 1Department of Information Security, Cryptography and Mathematics, Kookmin University, Seoul 02707, Republic of Korea; rladydwocjsw@kookmin.ac.kr; 2Department of Financial Information Security, Kookmin University, Seoul 02707, Republic of Korea; rptns777@kookmin.ac.kr

**Keywords:** covert channel, air gap, electromagnetic emission, smartphone security, silent audio playback, side-channel attack, data exfiltration

## Abstract

Covert channels enable hidden communication that poses significant security risks, particularly when smartphones are used as transmitters. This paper presents the first end-to-end implementation and evaluation of an electromagnetic (EM) covert channel on modern Samsung Galaxy S21, S22, and S23 smartphones (Samsung Electronics Co., Ltd., Suwon, Republic of Korea). We first demonstrate that a previously proposed method relying on zero-volume playback is no longer effective on these devices. Through a detailed analysis of EM emissions in the 0.1–2.5 MHz range, we discovered that consistent, volume-independent signals can be generated by exploiting the hardware’s recovery delay after silent audio playback. Based on these findings, we developed a complete system comprising a stealthy Android application for transmission, a time-based modulation scheme, and a demodulation technique designed around the characteristics of the generated signals to ensure reliable reception. The channel’s reliability and robustness were validated through evaluations of modulation time, probe distance, and message length. Experimental results show that the maximum error-free bit rate (bits per second, bps) reached 0.558 bps on Galaxy S21 and 0.772 bps on Galaxy S22 and Galaxy S23. Reliable communication was feasible up to 0.5 cm with a near-field probe, and a low alignment-aware bit error rate (BER) was maintained even for 100-byte messages. This work establishes a practical threat, and we conclude by proposing countermeasures to mitigate this vulnerability.

## 1. Introduction

Covert channels enable data exfiltration by exploiting unintended communication pathways, which pose significant security risks to isolated systems. This threat is particularly critical for smartphones, which store large volumes of sensitive user data and remain ubiquitous even in secure environments. Although prior studies have primarily examined smartphones as passive receivers, their potential role as active transmitters has received relatively little attention.

Previous work by [1] demonstrated the feasibility of an electromagnetic (EM) covert channel using emissions from a smartphone speaker. However, their study was limited to a proof-of-concept on older hardware and did not present a fully functional end-to-end system. Furthermore, our empirical evaluation shows that this technique fails on modern Samsung smartphones (Galaxy S21, S22, and S23, Samsung Electronics Co., Ltd., Republic of Korea), which no longer produce sustained EM emissions under the same conditions. This finding highlights a critical research gap.

To address this gap, we designed and implemented a complete system comprising a stealthy Android application for transmission, a time-based modulation scheme, and a demodulation method tailored to the characteristics of the generated signals. In addition, we conducted comprehensive performance evaluations to assess the reliability of the system in repeated transmissions and its robustness against variations in modulation time, probe distance, and message length. Our results confirm the feasibility of establishing an EM covert channel on modern smartphones, while also exposing important limitations in terms of transmission speed, effective range, and device diversity. These insights motivate further investigation into hardware-level constraints and potential techniques to improve practicality. Moreover, given the security implications, we discuss performance constraints in detail and propose countermeasures to mitigate the identified risks.

The primary contributions of this work are as follows:We perform the first detailed analysis of EM emissions from recent Samsung smartphones, identifying a new hardware-based mechanism that generates consistent, volume-independent signals.We design and implement a complete end-to-end covert channel system, including a stealthy Android transmitter application and a demodulation method tailored to signal characteristics.We rigorously evaluate the channel’s reliability and robustness through experiments on modulation time, probe distance, and message length, showing maximum error-free transmission rates of up to 0.772 bps, reliable communication up to 1 cm with a near-field probe, and low BER even for 100-byte messages.

The remainder of this paper is organized as follows. Section 2 reviews related work. Section 3 presents our analysis of smartphone EM emissions. Section 4 details the design of the proposed channel. Section 5 presents the performance evaluation. Section 6 discusses limitations and countermeasures, and Section 7 concludes the paper.

## 2. Related Work

Covert channels have emerged as a significant threat to air-gapped and secure networks, where conventional data transmission is intentionally restricted. These channels exploit unintended physical emissions, such as electromagnetic radiation [1,2,3,4,5,6,7], magnetic fields [8], acoustic signals [9,10,11], and power consumption patterns [12,13], from devices such as computers and smartphones to stealthily modulate and transmit data. Due to their covert nature and the potential to bypass traditional defenses, these channel attacks have attracted substantial attention in the security research community. In this section, we review three relevant areas of covert channel research: (1) electromagnetic transmission methods, (2) smartphone-based covert channels, and (3) smartphone-based electromagnetic covert channels.

### 2.1. Electromagnetic Covert Channels

Among various physical signal-based covert channels, those leveraging electromagnetic emissions have been extensively studied. For instance, AirHopper [2] encodes data by modulating video signals to generate FM radio emissions through the display cable. GSMem [3] induces electromagnetic emissions in the GSM frequency bands by simultaneously activating memory-related CPU instructions and utilizing the data buses between the CPU and memory. Bit-Jabber [4] constructs a covert channel by using frequent memory accesses to modulate the electromagnetic emissions generated by the DRAM clock, allowing data exfiltration even through concrete walls. USBee [5] achieves exfiltration by inducing electromagnetic emissions through USB data lines. More recently, AudioGap [6] has demonstrated that electromagnetic signals can be passively emitted from digital audio components such as sound cards by exploiting System Signals Auto Modulation (SSAM), using them as transmission media for covert data exfiltration. NoiseHopper [7] generates EM emissions by applying PWM through the digital output of the microcontroller.

Most existing electromagnetic covert channels are based on emissions from desktop-class hardware components. In contrast, the investigation of electromagnetic transmissions from smartphones remains limited, revealing a notable gap in the study of mobile-specific covert channels.

### 2.2. Smartphone-Based Covert Channels

Smartphones have predominantly been studied as receivers in covert communication systems. For instance, AirHopper [2] utilizes a smartphone’s FM receiver, MAGNETO [8] exploits its magnetic sensors, GAIROSCOPE [9] leverages gyroscope readings to decode transmitted signals, and Bit-Jabber [4] uses a rooted smartphone baseband processor.

Smartphones have also been used as transmitters. Notably BAT [10] and SonicEvasion [11] use inaudible ultrasonic signals for communication, whereas No Free Charge Theorem [12] and CovertPower [13] modulate smartphone power consumption to create side-channel transmissions. More recently, electromagnetic emissions from smartphones have been proposed as a potential medium for covert data transmission [1].

However, in most existing studies, smartphones function as passive receivers or transmit data using acoustic or power-based signals. The deliberate use of electromagnetic emissions from smartphones as a transmission mechanism remains largely unexplored.

### 2.3. Smartphone-Based Electromagnetic Covert Channel

In 2022, An et al. [1] proposed a covert channel that transmits data via electromagnetic emissions from a smartphone speaker during audio playback. The experiment used a Galaxy Note 8 device as the transmitter, playing a 300 ms silent audio file at a volume level of zero. During playback, the device emitted strong electromagnetic radiation centered at approximately 1.65 MHz. After the playback ended, the signal strength gradually decayed, with a notable drop occurring approximately 3000 ms later.

The method encoded a binary “1” by initiating a single playback event, sustaining electromagnetic emission for 3300 ms. A binary “0” was represented by silence for the same duration. The receiver captured these emissions in 3300 ms intervals and demodulated the signal using On–Off Keying (OOK), interpreting the high signal strength as “1” and the low signal strength as “0”.

However, this technique was evaluated only on older smartphones and required control over the volume setting. In addition, it only demonstrated the feasibility of the approach without implementing a fully functional covert channel in practice. This paper directly addresses these gaps by designing and implementing a practical, volume-independent implementation of an electromagnetic covert channel on recent Samsung smartphones. We demonstrate a complete end-to-end system that can be deployed in real-world air-gapped environments.

## 3. Electromagnetic Characteristics of Silent Audio Playback on Smartphones

In this section, we investigate the electromagnetic emissions generated during silent audio playback on three recent Samsung smartphones. This analysis highlights the limitations of a previously proposed covert channel technique and establishes a foundation for the development of a new electromagnetic-based covert channel. Section 3.1 details the experimental setup. Section 3.2 presents the shortcomings of the previous approach on recent Samsung smartphones. Section 3.3 examines the characteristics and origins of electromagnetic emissions observed in different smartphone models.

### 3.1. Experimental Setup

The experimental setup to analyze electromagnetic emissions generated during the audio playback of smartphones is shown in Figure 1. Component (a) is a Keysight EXA N9010B signal analyzer (Keysight Technologies, Santa Rosa, CA, USA), used to analyze the captured signals in the 10 Hz–44 GHz range. Component (b) is an LF-R 400 H-field probe (model 2-A; Langer EMV-Technik GmbH, Bannewitz, Germany), which captures electromagnetic emissions in the 100 kHz–50 MHz range. The probe was positioned approximately 0.1 cm from the bottom rear speaker area of each smartphone. Component (c) represents the three Samsung smartphones used for the experiments: Galaxy S21, S22, and S23, all of which are equipped with Class D audio amplifiers.

To facilitate the playback of silent audio, we developed a custom Android application in Kotlin using Android Studio. The application used the MediaPlayer class to repeatedly play silent audio files, and the intervals between successive playbacks were controlled using the delay function.

### 3.2. Failure of the Prior Covert Channel Method

We analyze electromagnetic emissions from Galaxy S21, S22, and S23 smartphones while playing a standard audio file at a volume level of zero for 5 s. As shown in Figure 2, a brief burst of electromagnetic activity was observed near 1.3 MHz, lasting approximately 0.1 s, after which no further emissions were detected. In addition, persistent background noise was present at various frequencies, but these signals were not related to audio playback. Due to the highly transient nature of the observed emission, the signal lacked the temporal consistency required for time-based OOK modulation. Consequently, the previously proposed technique was ineffective on newer smartphone models.

### 3.3. Electromagnetic Emissions During Silent File Playback

In this section, we investigate the characteristics of electromagnetic emissions during silent audio playback, focusing on volume dependence, signal re-emission timing, and device-specific behavior.

#### 3.3.1. Volume-Independent Electromagnetic Emissions

Previous work [1] focused exclusively on zero-volume audio playback. However, silent audio files, files that contain only silence, can be played regardless of the volume setting. When a 60 ms silent audio file was played on Samsung Galaxy S21, S22, and S23 devices with volume enabled, the resulting electromagnetic emissions closely matched those observed during standard playback at a volume level of zero. This indicates that the emissions originate not from the acoustic output itself but from the playback process, likely due to activity in the internal amplifier circuit. This observation is consistent with the fact that all three smartphones are equipped with Class D amplifiers, which are usually operated at switching frequencies in the 1.2–2 MHz range [14].

#### 3.3.2. Minimum Delay and Device Comparison for Signal Re-Emission

To examine the re-emission behavior of electromagnetic signals, we tested playback intervals of 2000 ms and 2300 ms on the Galaxy S22 device. No electromagnetic emissions were observed at 2000 ms, whereas a distinct re-emission was detected at 2300 ms. This result indicates that a minimum delay of approximately 2300 ms is required before the device can generate detectable electromagnetic signals again.

We further compared this behavior across three smartphone models: Galaxy S21, S22, and S23. As shown in Figure 3, Galaxy S21 required a longer delay of 3200 ms for signal reactivation, while S22 and S23 required only 2300 ms. This required delay for signal reactivation is consistent with the behavior of Class D audio amplifiers, which enter a power-saving mode during silence and require a certain recovery period before resuming normal switching activity leading to electromagnetic emissions [15].

#### 3.3.3. Identifying Usable Frequency Bands

We also examined which frequency bands were most suitable for signal detection. Figure 4 shows the distribution of the strength of the electromagnetic signal during silent playback in the three models. The 1.316–1.317 MHz frequency band exhibited relatively strong emissions with minimal ambient noise, making it a suitable candidate for demodulation. Among the three models, the Galaxy S21 produced the strongest signal, suggesting that signal pre-processing may be necessary for weaker emitting devices such as the Galaxy S23.

## 4. The Proposed Covert Channel

This section presents the design of a novel electromagnetic covert channel, based on the signal characteristics identified in Section 3. Section 4.1 introduces the threat and attack model. Section 4.2 describes the experimental communication setup. Section 4.3 explains the signal modulation technique. Finally, Section 4.4 describes the reception and noise removal strategies.

### 4.1. Attack Model

In this model, we consider an attacker attempting to exfiltrate sensitive information from an air-gapped environment, where neither Internet access nor conventional communication interfaces are available. A compromised smartphone, physically located within the secure zone, serves as the transmission medium.

The attacker first installs a malicious application on the smartphone, possibly using a physical infection vector such as a malicious USB charging station or cable, as demonstrated in previous work [16]. The application continuously plays a silent audio file. Although no audible sound is generated, this activity induces electromagnetic emissions from the internal audio amplifier circuit of the smartphone. These unintended signals are radiated and can be captured by a hidden probe or antenna placed close to each other, such as underneath a desk or embedded inside a phone stand.

The captured signals are then transmitted to the attacker’s receiving device (for example, a PC or embedded system) for demodulation and decoding, enabling exfiltration of covert data from the isolated environment. Figure 5 illustrates the overall attack model and the signal transmission pathway.

### 4.2. Communication Setup

The experimental setup for the proposed electromagnetic covert channel is shown in Figure 6. All experiments were conducted in a typical office environment with common ambient electromagnetic noise. Component (a) represents the receiving PC, which processes the digitized electromagnetic signals. Component (b) denotes the USRP N210 device (Ettus Research, Austin, TX, USA), which functions as an analog-to-digital converter (ADC) to digitize analog electromagnetic signals. Component (c) indicates the transmitting smartphone. Component (d) shows the position of the near-field H-field probe, placed to detect emissions from the speaker area of smartphones.

When playing a silent audio file on the smartphone, the internal Class D amplifier emits electromagnetic signals, which are captured by an LF-R 400 H-field probe (model 2-A) positioned near the speaker. The captured analog signals were digitized using a USRP N210 ADC and forwarded to the receiving PC. On the PC, GNU Radio was employed to filter and extract only the signals in the 1.316–1.317 MHz frequency band. To reduce noise and improve measurement stability, an fast Fourier transform (FFT)-based averaging technique was applied, in which the spectral outputs were averaged over each FFT size block. The resulting signals were then demodulated and analyzed. The detailed GNU Radio configuration parameters are summarized in Table 1, which correspond to the values at which the signals were clearly visible above the noise floor during the experiments.

To implement covert transmission, we developed a custom Android application using Kotlin in Android Studio. To maintain stealth, the application is designed to appear as a benign search tool, as shown in Figure 7a, where it is disguised as an icon on the home screen. When launched, the app presents a simple user interface that resembles a typical search bar (Figure 7b). Internally, it utilizes the MediaPlayer class to repeatedly play a short silent audio file. The user specifies three input parameters on the interface: the transmission delay for bit “1,” the delay for bit “0,” and a payload expressed in hexadecimal format (e.g., “3200 500 ab6e”). The application converts the hexadecimal payload into a binary sequence and transmits each bit by controlling the playback delay accordingly, thereby implementing timing-based modulation.

This design allows the generation of unintended electromagnetic emissions from the smartphone audio subsystem without requiring root access or additional hardware. By embedding the covert transmission logic in an ordinary-looking mobile app, our approach demonstrates a practical, stealthy, and mobile-originated electromagnetic covert channel. A search tool was chosen as a disguise because its functionality does not raise suspicion, and user input into a search bar provides a natural pretext to initiate data encoding and transmission. This highlights the feasibility of smartphone-based transmission in real-world scenarios, filling a critical gap in existing research, which primarily focuses on desktop-class hardware as transmission sources.

### 4.3. Signal Modulation

We propose a modulation scheme based on timing, which takes advantage of two key properties: (1) electromagnetic signals are emitted during silent audio playback, and (2) a minimum delay is required between playbacks to ensure signal re-emission. Time-based modulation has been widely explored in covert communication [17]. Our work leverages timing as a modulation dimension, while applying it in the context of electromagnetic leakage from smartphones. The detailed encoding process is described in Algorithm 1. Each bit in the input bit sequence is encoded according to its value. For each “1” bit, the system plays a silent audio file to trigger electromagnetic emissions, followed by a waiting period corresponding to the minimum delay time (3200 ms for Galaxy S21, or 2300 ms for Galaxy S22 and S23). For each “0” bit, the system simply waits for a fixed interval, referred to as the “zero-bit delay,” which can be adjusted to optimize transmission latency. This encoding process is repeated for the entire bit sequence, with an initial transmission of a “1” bit to mark the start of the transmission, ensuring that the received sequence can be decoded accurately, even if the transmitted sequence starts with a “0.” After the sequence ends, one final silent file playback is triggered to mark the end of the transmission.

The receiver decodes the transmitted message by measuring the time intervals between successive electromagnetic emissions. For example, if the zero-bit delay is set to 500 ms on the Galaxy S21, then a measured interval of 3200 ms between peaks represents a single “1” bit. Consequently, an interval of 3700 ms (3200 ms + 500 ms) would represent “10”, 4200 ms would represent “100”, etc. An example of this time-based modulation scheme is illustrated in Figure 8.
**Algorithm 1** Signal modulation**Input:** bit_sequence, minimum_delay_time, zero_bit_delay**Output:** modulated_signal1:play(silent_sound)                                                             ▹ Start with the initial “1” bit2:delay(minimum_delay_time)3:**for** i=0 to len(bit_sequence)-1 **do**4:    **if** bit_sequence[*i*] == 1 **then**5:        play(silent_sound)6:        delay(minimum_delay_time)                                     ▹ Delay required for “1” bit7:    **else**8:        delay(zero_bit_delay)                                                       ▹ Fixed delay for “0” bit9:    **end if**10:**end for**11:play(silent_sound)                     ▹ End the transmission with a final silent playback

### 4.4. Signal Demodulation

Figure 9 shows the signals received on the PC after transmitting messages from three smartphones using the covert channel application, with the probe placed at a distance of 0.1 cm. As analyzed in Section 3.3, the signal characteristics vary significantly between models. For Galaxy S21 (Figure 9a), the signal is strong and clean, allowing for straightforward demodulation by applying a fixed threshold and measuring the time intervals between the peaks. In contrast, Galaxy S22 (Figure 9b) and S23 (Figure 9c) exhibit weaker amplitudes and intermittent noise between the main signal peaks. This noise makes a simple threshold-based approach unreliable, as the noise can be mistaken for a valid signal peak. To address this issue and enable robust demodulation across all devices, we introduce Algorithm 2.

The Algorithm 2 reconstructs the transmitted bit sequence by analyzing the time intervals between consecutive electromagnetic peaks in the received signal. The algorithm first initializes a counter to measure the number of samples between peaks. When a sample exceeds the detection threshold, it marks the start of a new interval. If another peak is detected after at least “1” bit length (*l*), the algorithm computes the interval Δ between the two peaks. Based on Δ, it estimates the number of zero bits that occurred between the two “1” bits. Specifically, the number of zeros is calculated as k=⌊(Δ−l)/z⌋, where *z* denotes the nominal length of a “0” bit. The algorithm then appends a “1” followed by *k* zeros to the reconstructed sequence. This process is repeated for the entire signal, thereby converting a noisy peak trace into the corresponding bit sequence. By explicitly modeling the minimum delay of “1” bits and the relative timing of “0” bits, the algorithm achieves robust demodulation even in the presence of noise between the peaks.
**Algorithm 2** Demodulation**Input:** *s* (signal), *l* (1-bit length), *z* (0-bit length), *t* (threshold)**Output:** recovered_sequence1:Initialize count←0, recovered_sequence ←∅2:**for** each sample *x* in *s* **do**3:    **if** x≥t and count=0 **then**4:        Start new interval: count←15:    **else if** x≥t and count≥l **then**6:        Compute interval Δ←count7:        Estimate number of “0” bits: k←⌊(Δ−l)/z⌋8:        Append “1” followed by *k* zeros to recovered_sequence9:        Reset count←110:    **else if** count>0 **then**11:        Increment count12:    **end if**13:**end for**14:**return** recovered_sequence

## 5. Performance Evaluation

Following the finite-length evaluation perspective highlighted in [18], we evaluated the performance of the proposed covert channel using messages of limited length on three Samsung smartphones (Galaxy S21, S22, and S23), rather than relying on asymptotic assumptions. The evaluation is composed of three parts: the effect of the zero-bit delay parameter (Section 5.2), reception performance at different probe distances (Section 5.3), and message-length effects (Section 5.4). To exercise realistic message structures in the first two parts (Section 5.2 and Section 5.3), we transmit representative 10-byte sequences generated as random hexadecimal strings, where each bit is independently drawn with equal probability of being “0” or “1.” For the third part (Section 5.4), we use 50-byte and 100-byte random hexadecimal strings generated under the same conditions to study length effects. For each experimental condition, five different random hexadecimal strings were transmitted and the reported results represent the average of these trials. Together, these three experiments allow us to quantify both the reliability and throughput of the proposed scheme under realistic conditions. All evaluations are conducted using the metrics defined in Section 5.1.

### 5.1. Metrics

To evaluate the performance of the proposed covert channel, we consider two key metrics: the alignment-aware bit error rate, denoted ${\mathrm{BER}}_{\mathrm{align}}$, and the bit rate in bits per second (bps).

Since the proposed covert channel is based on timing modulation, the conventional Hamming distance bit error rate is insufficient because it does not capture insertion and deletion errors. We therefore adopt ${\mathrm{BER}}_{\mathrm{align}}$, defined as(1)$${\mathrm{BER}}_{\mathrm{align}}=\frac{N_{\mathrm{ins}}+N_{\mathrm{del}}+N_{\mathrm{sub}}}{N_{\mathrm{orig}}}.$$ Here, $N_{\mathrm{orig}}$ is the number of bits in the original transmitted message, $N_{\mathrm{ins}}$ is the number of inserted bits, $N_{\mathrm{del}}$ is the number of deleted bits, and $N_{\mathrm{sub}}$ is the number of substituted bits. The error counts are obtained through sequence alignment (Levenshtein distance), which accounts for all types of errors in finite-length transmissions.

The bit rate (bps) represents the transmission speed of the covert channel and is defined as(2)$$\mathrm{bps}=\frac{N_{\mathrm{orig}}}{T_{\mathrm{tx}}}.$$
where $N_{\mathrm{orig}}$ is the total number of transmitted bits and $T_{\mathrm{tx}}$ is the overall transmission time in seconds.

### 5.2. Zero-Bit Delay Optimization

We first evaluated the effect of the zero-bit delay parameter on system performance, with the probe distance fixed at 0.1 cm for all experiments. The average bit rate (bps) and ${\mathrm{BER}}_{\mathrm{align}}$ were measured using five different 10-byte random hexadecimal messages, and the reported values represent the average across these trials, for zero-bit delay values of 50 ms, 100 ms, 200 ms, 300 ms, 400 ms, and 500 ms.

Table 2 and Figure 10 present the ${\mathrm{BER}}_{\mathrm{align}}$ results. At 400 ms and 500 ms, all devices achieved ${\mathrm{BER}}_{\mathrm{align}}=0$, confirming reliable performance under these conditions. For Galaxy S21, ${\mathrm{BER}}_{\mathrm{align}}$ remained very low down to 100 ms but showed a noticeable increase at 50 ms. Galaxy S22 was more sensitive, as ${\mathrm{BER}}_{\mathrm{align}}$ was 0 at 300 ms and remained relatively low at 200 ms, but rose sharply at 100 ms and became high at 50 ms, indicating weaker tolerance to short delays. Galaxy S23 reached ${\mathrm{BER}}_{\mathrm{align}}=0$ already at 300 ms, maintained very low ${\mathrm{BER}}_{\mathrm{align}}$ at 200 ms and 100 ms, and showed only a slight increase at 50 ms. Consequently, Galaxy S22 was the most sensitive to reduced delays, while Galaxy S23 was the least affected, exhibiting the most robust performance among the three devices.

In addition, Table 3 and Figure 11 summarize the performance results (bps) for the same set of messages. Galaxy S21 consistently achieved the lowest bit rate because the time required for the modulation of a “1” bit was longer. In contrast, Galaxy S22 and Galaxy S23 exhibited identical bit rate values, as both devices shared the same modulation time for the “1” bit. As shown in the ${\mathrm{BER}}_{\mathrm{align}}$ results, Galaxy S23 showed the best performance when the probe was placed at a distance of 0.1 cm. Its low ${\mathrm{BER}}_{\mathrm{align}}$ and high bps demonstrate that it can effectively distinguish between the “1” and “0” bit intervals, making it the optimal choice for the given setup.

### 5.3. Probe Distance Evaluation

Table 4 and Figure 12 present the ${\mathrm{BER}}_{\mathrm{align}}$ results measured at different probe distances. For this evaluation, the zero-bit delay was fixed to the smallest value that achieved ${\mathrm{BER}}_{\mathrm{align}}=0$ for each device, namely 400 ms for Galaxy S21 and 300 ms for Galaxy S22 and Galaxy S23, as determined in Section 5.2.

The results show that performance closely followed the measured signal strength. At a distance of 0.1 cm, all devices transmitted without errors. At 0.5 cm, Galaxy S22 and Galaxy S23 began to show small but noticeable error rates, while Galaxy S21 maintained error-free transmission. At 1 cm, the signals from Galaxy S22 and Galaxy S23 became indistinguishable from the noise, making demodulation impossible. In contrast, Galaxy S21 was still able to transmit reliably at 1 cm, consistent with the signal strength measurements reported in Section 3, which showed that Galaxy S21 produced the strongest emissions. However, at 1.5 cm, even Galaxy S21 exhibited deterioration in performance as the signal was no longer clearly separable from background noise.

These results indicate that with the near-field probe used in our experiments, reliable transmission was feasible up to approximately 1 cm only on the Galaxy S21. In contrast, the signals from the Galaxy S22 and S23 became indistinguishable from noise beyond 0.5 cm. With more sensitive far-field antenna equipment, the effective transmission range could potentially be extended.

### 5.4. Message Length Evaluation

We conducted additional experiments to evaluate the performance with longer message lengths and verify the robustness of the proposed system under extended transmissions. Long transmissions may be influenced by factors such as device heating or background processes on smartphones. To test this, we used 10-byte, 50-byte, and 100-byte random hexadecimal messages at a fixed probe distance of 0.1 cm. For each condition, five different random messages were transmitted and the reported values represent the mean ${\mathrm{BER}}_{\mathrm{align}}$ in these trials.

Table 5 and Figure 13 show the results. Galaxy S21 and Galaxy S22 achieved near-zero ${\mathrm{BER}}_{\mathrm{align}}$ even for the longest 100-byte messages, demonstrating stable and similar performance. Galaxy S23 also showed low ${\mathrm{BER}}_{\mathrm{align}}$, but was more affected by message length due to its relatively weaker signal strength, making it more susceptible to noise during longer transmissions.

Overall, all three devices maintained a low ${\mathrm{BER}}_{\mathrm{align}}$ under extended message lengths, confirming the robustness of the proposed system. In contrast, Galaxy S21 and S22 demonstrated almost identical resilience, whereas Galaxy S23 was slightly more affected by longer transmissions, although its ${\mathrm{BER}}_{\mathrm{align}}$ remained sufficiently low to support reliable communication.

## 6. Discussion and Countermeasures

Our work not only demonstrates a practical electromagnetic covert channel on modern smartphones but also reveals several limitations and areas for future improvement. In this section, we discuss the performance constraints of the proposed channel and present potential countermeasures to mitigate these threats.

### 6.1. Limitations and Future Work

The proposed covert channel achieves a low bit error rate, demonstrating the reliability of the method. However, it has limitations in terms of transmission speed and operating distance. As shown in Table 3, reducing the zero-bit delay does not significantly increase the bit rate because the time to transmit the “1” bit remains relatively long due to hardware constraints. Future work could focus on improving the transmission rate by investigating software-level methods to prevent the speaker’s Class D amplifier from entering its power-saving mode, which would eliminate the long re-emission delay.

With respect to distance, our experiments were conducted using a near-field H-field probe. The effective range was limited to approximately 0.5 cm, as signals became too weak for reliable demodulation beyond this point. We believe that with more sensitive, far-field antenna equipment, the transmission range could be extended, making the attack more practical in real-world scenarios.

Finally, our experiments were limited to Samsung smartphones. Preliminary observations on Apple’s iPhone models suggest that electromagnetic emissions are also present during silent audio playback, although the signal strength is significantly weaker. This difference may arise from variations in amplifier circuitry or internal hardware shielding. Future work will explore whether this technique can be adapted to other smartphone brands, possibly through external amplification or advanced pre-processing techniques.

### 6.2. Countermeasures

To mitigate the risks posed by smartphone-based electromagnetic covert channels, several countermeasures can be implemented across different layers of security:**Software-level restrictions.** Restrict or monitor access to low-level audio playback APIs, such as SoundPool or AudioTrack, especially in secure environments. Operating systems could implement permission models that alert users or administrators when an application repeatedly accesses audio hardware without producing audible sound.**Environmental scanning.** Regularly scan the environment using spectrum analyzers or dedicated electromagnetic probes to detect and investigate unexpected emissions in critical frequency ranges. An established baseline of the ambient EM environment can help identify anomalous signals indicative of a covert channel.**Air-gap enforcement.** Enforce strict physical and logical separation policies. This includes banning personal or unauthorized electronic devices from air-gapped zones to eliminate potential transmission vectors. Although challenging to enforce, this remains one of the most effective defenses against physical-layer covert channels.

## 7. Conclusions

This paper presents the first end-to-end implementation and evaluation of an electromagnetic covert channel on modern Samsung Galaxy S21, S22, and S23 smartphones. We started by analyzing the unique EM emissions of each device, discovering that the previous method [1] was ineffective. Based on these findings, we developed a complete system: a stealthy Android application for transmission, a time-based modulation scheme, and a demodulation method tailored to the characteristics of the generated signals.

The reliability of the channel was validated through three evaluations that focused on the zero-bit delay, probe distance, and message length. Error-free transmission was achieved with a minimum zero-bit delay of 400 ms on the Galaxy S21 and 300 ms on the Galaxy S22 and S23, corresponding to bit rates of 0.558 bps and 0.772 bps, respectively. Reliable communication was possible up to about 1 cm with the near-field probe, and the Galaxy S21 performed best at longer distances. In addition, all devices maintained a low BER even for 50-byte and 100-byte messages, confirming that the proposed system remains robust under extended transmission durations.

By successfully demonstrating a complete and functional covert channel on modern smartphones, this study goes beyond theoretical feasibility and establishes a validated practical threat. To mitigate this vulnerability, we conclude by proposing several physical and software-level countermeasures.

## Figures and Tables

**Figure 1 sensors-25-05900-f001:**
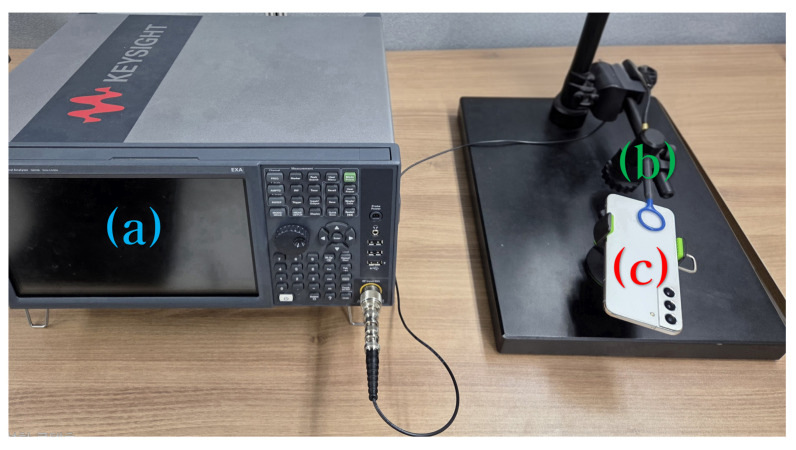
Experimental setup for electromagnetic (EM) emission characterization. (**a**) Keysight EXA-N9010B signal analyzer, used to analyze captured signals in the 10 Hz–44 GHz range. (**b**) LF-R 400 H-field probe (model 2-A), positioned approximately 0.1 cm from the smartphone’s rear speaker to capture EM emissions in the 100 kHz–50 MHz range. (**c**) Three Samsung smartphones (Galaxy S21, S22, and S23), each equipped with a Class D audio amplifier for playback experiments.

**Figure 2 sensors-25-05900-f002:**
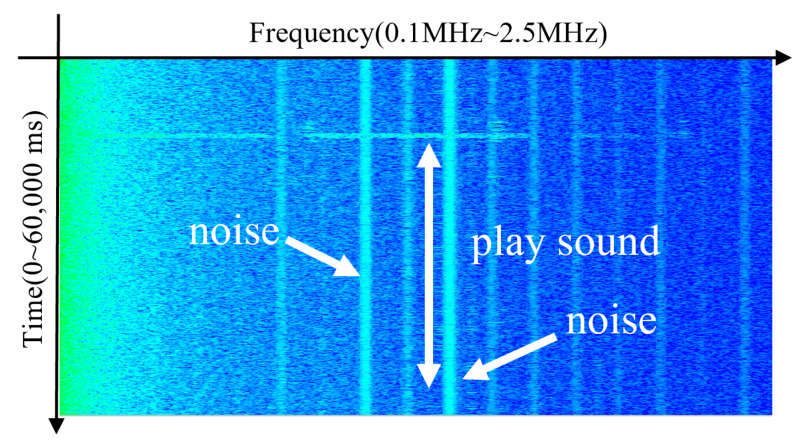
Electromagnetic signal emissions during played sound at a volume level of zero for 5 s on the Galaxy S22.

**Figure 3 sensors-25-05900-f003:**
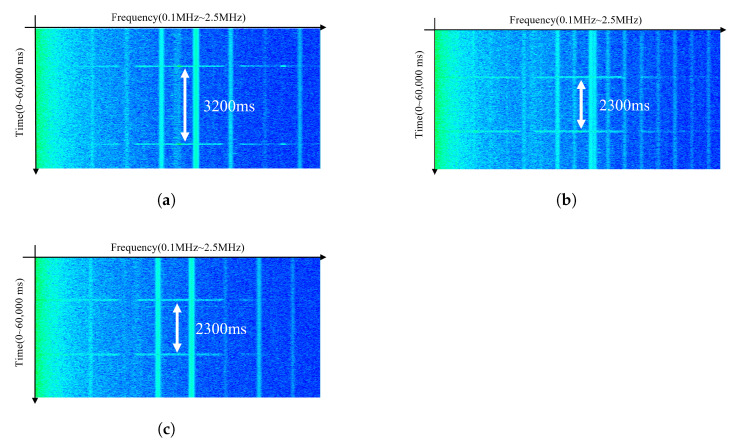
Delay-dependent electromagnetic signal re-emission characteristics for three smartphone models. A minimum delay is required for signal reactivation: (**a**) Galaxy S21 (3200 ms). (**b**) Galaxy S22 (2300 ms). (**c**) Galaxy S23 (2300 ms).

**Figure 4 sensors-25-05900-f004:**
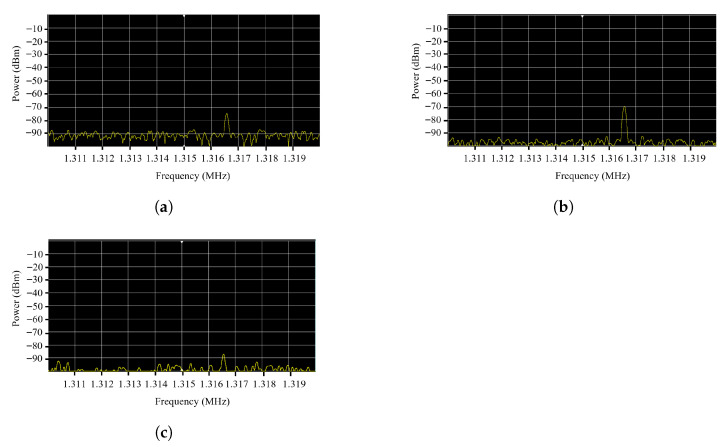
Usable frequency band analysis for three smartphone models: (**a**) Galaxy S21. (**b**) Galaxy S22. (**c**) Galaxy S23.

**Figure 5 sensors-25-05900-f005:**
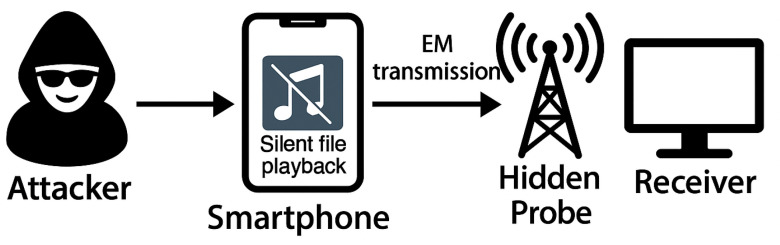
Electromagnetic signal transmission from a compromised smartphone in an air-gapped environment. Silent audio playback generates emissions that are captured by a hidden probe and replayed to the attacker’s receiver.

**Figure 6 sensors-25-05900-f006:**
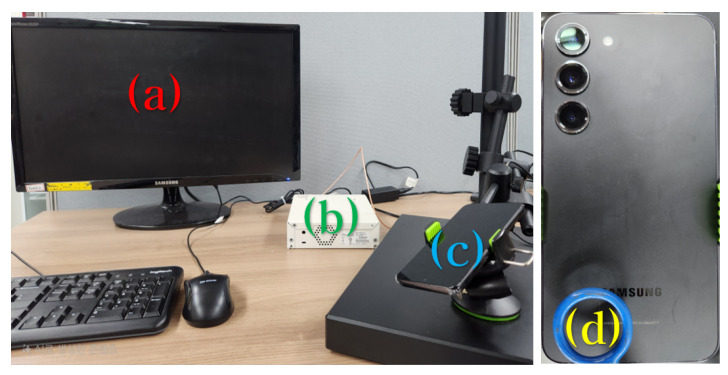
Experimental setup of the smartphone-based electromagnetic covert channel. (**a**) Receiving PC, which processes the digitized electromagnetic signals. (**b**) USRP N210 device, functioning as an analog-to-digital converter (ADC) to digitize analog EM signals. (**c**) Transmitting smartphone used to generate EM emissions. (**d**) Near-field H-field probe positioned at the smartphone’s speaker area to capture the emissions.

**Figure 7 sensors-25-05900-f007:**
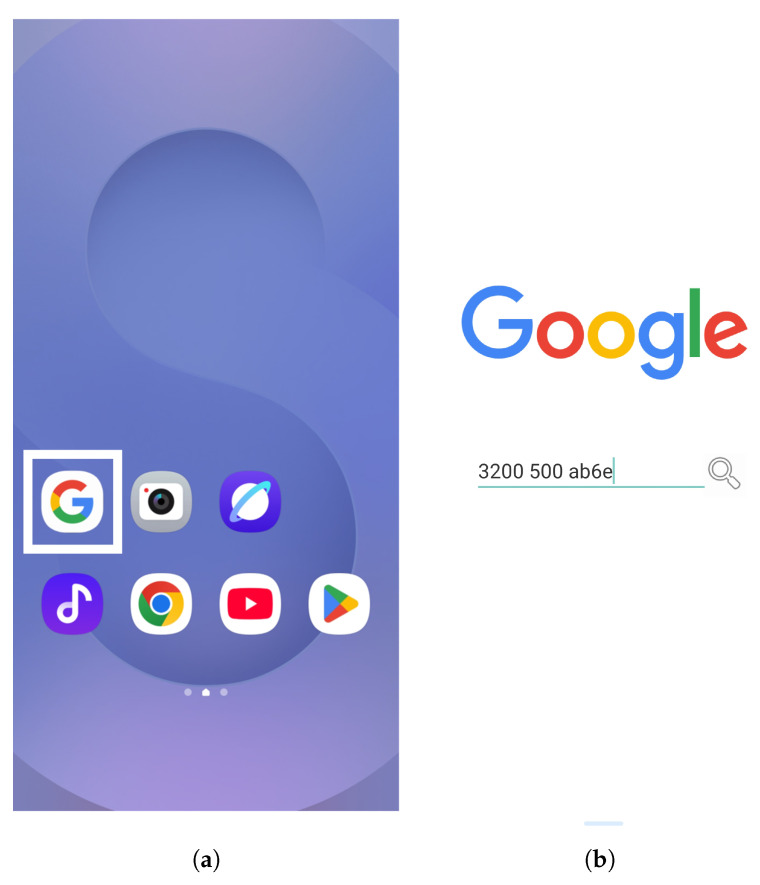
Interface of the transmitter application developed in Kotlin: (**a**) App icon disguised as a search tool on the home screen. (**b**) Main user interface of the transmitter app, where the user specifies the transmission delay for bit “1,” the delay for bit “0,” and a payload in hexadecimal format, which is then converted into a bit sequence for covert transmission.

**Figure 8 sensors-25-05900-f008:**
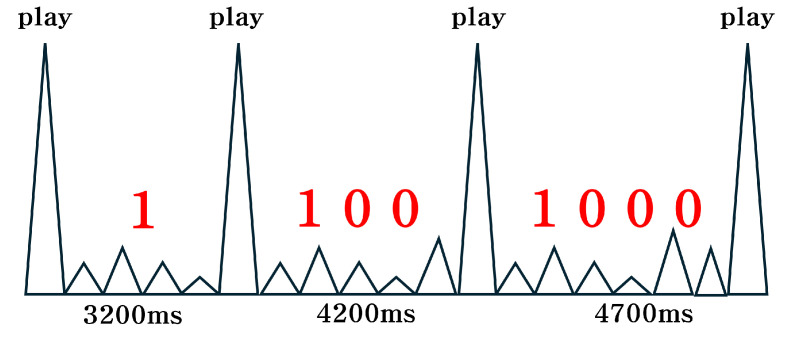
Example of time-based electromagnetic signal modulation on Galaxy S21, where a zero-bit delay of 500 ms is used and longer intervals correspond to different binary values.

**Figure 9 sensors-25-05900-f009:**
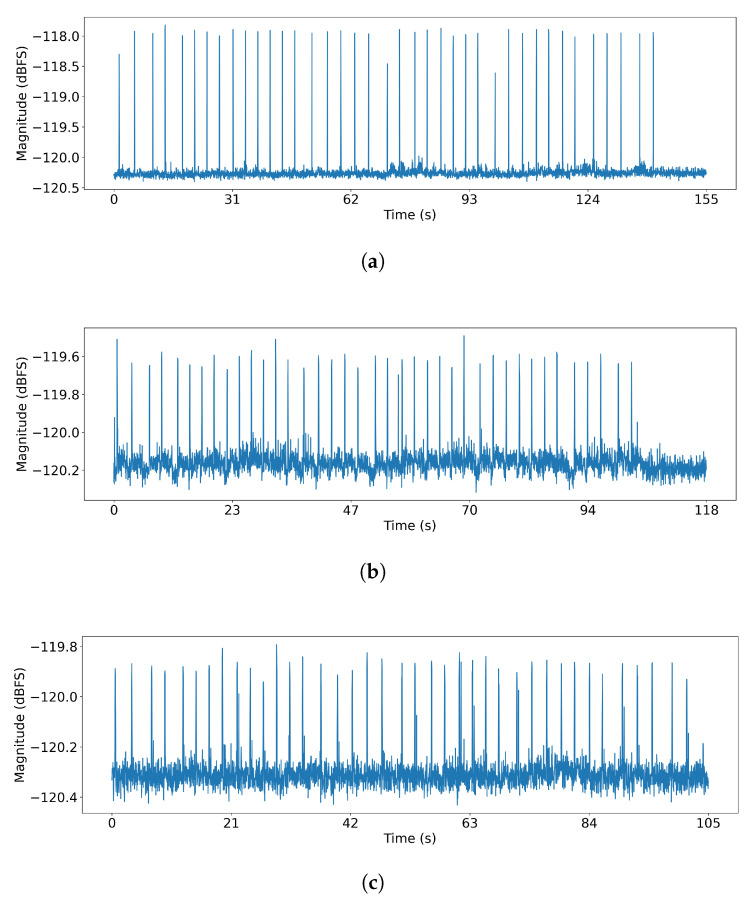
Received electromagnetic signals with the probe placed at a distance of 0.1 cm: (**a**) Galaxy S21. (**b**) Galaxy S22. (**c**) Galaxy S23.

**Figure 10 sensors-25-05900-f010:**
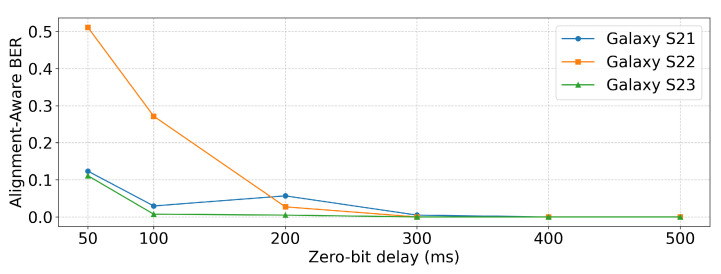
${\mathrm{BER}}_{\mathrm{align}}$ performance under different zero-bit-delay values.

**Figure 11 sensors-25-05900-f011:**
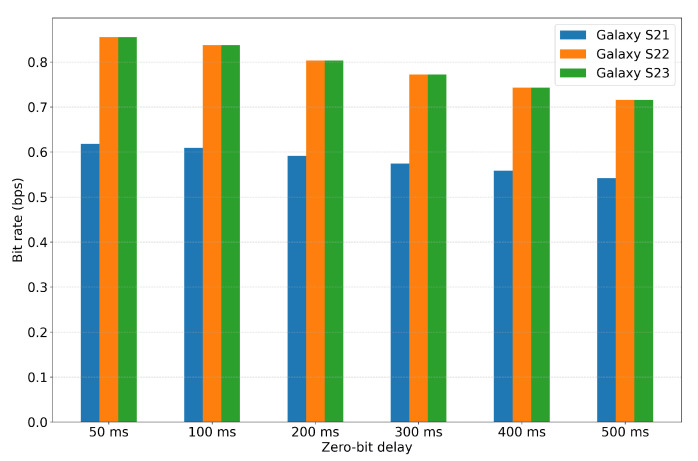
Bit rate (bps) performance under different zero-bit-delay values.

**Figure 12 sensors-25-05900-f012:**
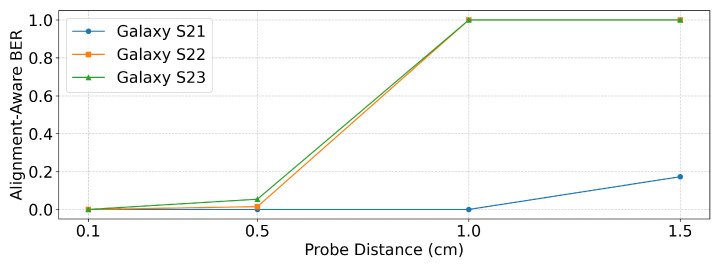
${\mathrm{BER}}_{\mathrm{align}}$ performance under different probe distances.

**Figure 13 sensors-25-05900-f013:**
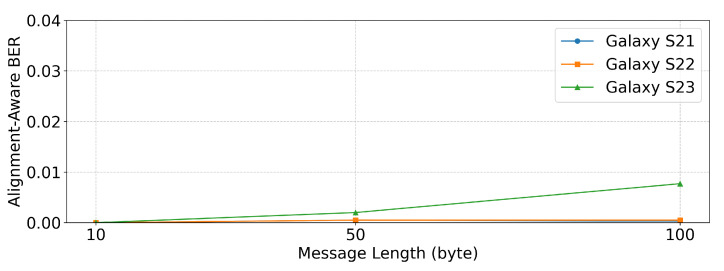
${\mathrm{BER}}_{\mathrm{align}}$ performance under different message lengths.

**Table 1 sensors-25-05900-t001:** GNU Radio configuration for spectrum acquisition and demodulation.

Parameter	Value
Center frequency	$f_{c}=1.3165\;\mathrm{MHz}$
Channel bandwidth	B=1kHz
FFT size	$N_{\mathrm{FFT}}=65,536$
Frame rate	$f_{\mathrm{frame}}=30\;\mathrm{frames}/s$
Sample rate	$f_{s}=1,966,080\;S/s$ (≈1.966MS/s)

**Table 2 sensors-25-05900-t002:** Alignment-aware bit error rate (${\mathrm{BER}}_{\mathrm{align}}$) under different zero-bit delay values.

Device	50 ms	100 ms	200 ms	300 ms	400 ms	500 ms
Galaxy S21	0.1235	0.0296	0.0568	0.0049	0	0
Galaxy S22	0.5111	0.2716	0.0272	0	0	0
Galaxy S23	0.1111	0.0074	0.0049	0	0	0

**Table 3 sensors-25-05900-t003:** Bit rate (bps) results under different zero-bit-delay values.

Device	50 ms	100 ms	200 ms	300 ms	400 ms	500 ms
Galaxy S21	0.618	0.609	0.591	0.574	0.558	0.542
Galaxy S22	0.855	0.837	0.803	0.772	0.743	0.716
Galaxy S23	0.855	0.837	0.803	0.772	0.743	0.716

**Table 4 sensors-25-05900-t004:** ${\mathrm{BER}}_{\mathrm{align}}$ results under different probe distances.

Device	0.1 cm	0.5 cm	1 cm	1.5 cm
Galaxy S21	0	0	0	0.1728
Galaxy S22	0	0.0148	1	1
Galaxy S23	0	0.0543	1	1

**Table 5 sensors-25-05900-t005:** ${\mathrm{BER}}_{\mathrm{align}}$ results under different message lengths.

Device	10 bytes	50 bytes	100 bytes
Galaxy S21	0	0.0005	0.0003
Galaxy S22	0	0.0005	0.0005
Galaxy S23	0	0.0020	0.0077

## Data Availability

The traces used in our demonstrations are available at the following link: https://drive.google.com/file/d/1KDI43sUBQxVakj_xQxoiKnLV-LRDMFsO/view?usp=sharing (accessed on 11 September 2025).

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
