# Peer review of "EMPhone: Electromagnetic Covert Channel via Silent Audio Playback on Smartphones"

_sensors, 2025, doi:10.3390/s25185900_

Round 1
Reviewer 1 Report
Comments and Suggestions for Authors
The article presents an interesting concept of an air-gap channel for smartphones. The authors correctly identified the vulnerability, analyzed its behavior, and designed a proof-of-concept solution.
However, I have concerns about the study:
1. How was the "0" bit length of 500 ms determined? Could a shorter duration have been established?
2. Why was "10000000001" chosen as the preamble? Why is it so long? Wouldn't multiple consecutive zeros cause desynchronization?
3. Was the bit rate calculated with or without the preamble?
4. I have the impression that the designed message framing protocol is not necessary for measuring BER. This protocol appears to be far from optimal.
5. How many trials were performed in the study? One for each phone model? This gives the impression of the cherry-picking.
About the figures:
1. Figure 1 - the image is too small, illegible, lacks explanation of symbols (a), (b), (c)
2. Figure 4 - the vertical axes are described as "Log scale" but what is the physical quantity? Power?
3. Figure 9 - are all vertical axes really power? What units are they?
The article is interesting, but the experiment is not representative. For this reason, the article is more suitable for a technical blog. It is necessary to conduct a larger number of experiments on random data, not on one fixed message. I suggest abandoning the framing protocol, establishing one fixed message length, and performing multiple repetitions of transmissions of different messages of the same length. Otherwise, the article should not be published.
Reviewer 2 Report
Comments and Suggestions for Authors
The research presents a novel approach to exploiting electromagnetic emissions from smartphones for covert communication, demonstrating feasibility on modern Samsung devices. The reviewer has the following questions
1. Given the 0.3 cm probe distance requirement in Figure 1, what maximum practical distance achieves reliable signal detection in real-world environments, and how rapidly does BER degrade with increasing distance?
2. Why was the 500 ms "0"-bit delay selected instead of empirically minimizing it to maximize transmission rate, and what fundamental hardware limitation prevents further reduction?
3. How does BER scale when transmitting payloads exceeding the 31-byte limit, particularly under sustained transmission where thermal effects or background processes may alter emission characteristics?
4. For proposed EMI shielding countermeasures in Section 4.7, what attenuation levels (dB) were experimentally measured when applying standard conductive materials to speaker assemblies?
5. Suggest adding some new relevant references [R1] - [R2] in the field of covert communication to enrich the background and support analysis.
[R1] “Covert mmWave communications with finite blocklength against spatially random wardens, ” IEEE Internet of Things Journal, vol. 11, no. 2, pp. 3402–3416, Jan. 2024.
[R2] “Improving Age of Information for Covert Communication with Time-Modulated Arrays,” IEEE Internet of Things Journal, vol. 12, no. 2, pp. 1718-1731, Jan. 2025.
Round 2
Reviewer 1 Report
Comments and Suggestions for Authors
Thank you for the changes. The article has significantly changed in form and content. However, there are still imperfections that need to be corrected.
Algorithm 3 (4.4.2) has been added, which constitutes a significant qualitative change compared to the previous version of the article. This algorithm uses the parameters: l, p, and d. How were their values determined? How does changing the values of these parameters affect the BER? This is not explained in the current version of the article, and it is not possible to reproduce the results of experiments.
Section 5.2, "Zero-Bit Delay Optimization" describes the method used to determine the duration of the 0-bit. The BER value was analyzed for different time durations. However, I wonder why the values of 50, 100, 200, and 500 ms were chosen. The analyzed times are unevenly distributed, and there is a very clear lack of experiments for time values in the range between 200 and 500 ms. The results in Table 1 are an attempt to justify the pre-assumed a priori value of 500 ms, which authors did not justify! The study must be repeated for values such as 250, 300, 350, 400, and 450 ms. I also suggest presenting the results in the form of a graph.
In sections 5.2 and 5.3, five words were used in the experiments: conference, themselves, literature, government, everything. The choice of ASCII strings for this research is a mistake. ASCII strings composed of lowercase English letters have values ranging from 97 to 122, and these values fall within the seven least significant bits of the eight bits used. Furthermore, for these values, two bits (6 and 7) are always constant and have a value of 1. A brief analysis shows that in these words, bit 1 occurs about 20% more frequently than bit 0. The encoding method used, where a zero is encoded by a specifically long period of no signal, favors strings with more ones than zeros. This is cherry-picking, and the research must be repeated for strings where the probability of bit 0 and bit 1 occurrence is equal.
How many experiments were performed in the study in section 5.4? The authors only mention using 50 and 100-byte random text strings. How many such random text strings were transmitted? Were they still ASCII character strings? What was their entropy value?
The studies presented in sections 5.2, 5.3, and 5.4 need to be repeated.
Comments on the figures:
1. Figure 1. The notations a, b, and c should be briefly explained in the figure's caption, not just in the paragraph on the previous page. The same applies to Figure 6.
2. Figure 4. The decibel (dB) is a relative unit. If a reference value has not been defined, this is a significant error. Please use unambiguous units on the scale (e.g., dBm). The same applies to Figure 9, Figure 10, and Figure 11.
Reviewer 2 Report
Comments and Suggestions for Authors
All comments have been addressed satisfactorily, and the reviewer deems the manuscript suitable for publication.
Author Response
We would like to thank the reviewer for taking the time to read thoroughly the paper, for the precise suggestions he made to improve it and for all the errors they brought to our attention. In revision, we have done our best to address the issues raised by the reviewer as much as possible.
Importantly, all experiments were repeated using randomly generated hexadecimal strings, and the results of every experiment are now presented both in tables and in graphical form for clarity.
Thank you very much for taking the time to review this manuscript. We highly appreciate your valuable feedback and have carefully considered all of the comments provided. Based on your suggestions, we have made several revisions to improve the manuscript.
- We re-examined Algorithm 3 and found that its parameters had to be manually tuned. Since adjusting GNU Radio parameters has already improved the signal-to-noise separation significantly, Algorithm 3 did not provide further benefit. Therefore, it was removed from the revised manuscript. The related setup parameters are now clearly summarized in Section 4.2 (Table 1).
- We repeated the zero-bit delay experiments with expanded values (50, 100, 200, 300, 400, 500 ms) and added graphical results. This clarified the performance trends and confirmed optimized settings: 400 ms for Galaxy S21 and 300 ms for Galaxy S22/S23. Results are reported in Section 5.2 (Table 2, Figure 10).
- We replaced ASCII test words with unbiased random hexadecimal strings generated using Python’s random module, ensuring equal probability of 0s and 1s. All experiments in Sections 5.2–5.4 were repeated with five independent random strings per condition, and results are now averaged across trials.
- For the message length evaluation (Section 5.4), we extended the design by testing with five distinct 50-byte and 100-byte random strings, rather than a single long ASCII message. This ensures balanced entropy (1 bit per bit) and more reliable evaluation.
- Figure captions were revised to include explicit explanations of notations (a), (b), (c) for Figures 1 and 6.
- Axis labels in Figures 4 and 9 were corrected to clearly distinguish between absolute (dBm) and relative (dBFS) units.
We believe these changes address the reviewer’s concerns and strengthen the manuscript. The updated version now provides clearer methodology, expanded experimental coverage, and more rigorous validation of the proposed covert channel system.